# Sub-3 Å Cryo-EM Structures of Necrosis Virus Particles via the Use of Multipurpose TEM with Electron Counting Camera

**DOI:** 10.3390/ijms22136859

**Published:** 2021-06-25

**Authors:** Chun-Hsiung Wang, Dong-Hua Chen, Shih-Hsin Huang, Yi-Min Wu, Yi-Yun Chen, Yeukuang Hwu, David Bushnell, Roger Kornberg, Wei-Hau Chang

**Affiliations:** 1Institute of Chemistry, Academia Sinica, Nangang, Taipei 11529, Taiwan; clarinetwang1@gmail.com (C.-H.W.); robhsh@gmail.com (S.-H.H.); marinesean@gmail.com (Y.-M.W.); 2Structural Biology, School of Medicine, Stanford University, Stanford, CA 94305, USA; donghua.chen@gmail.com (D.-H.C.); bushnell@stanford.edu (D.B.); kornberg@stanford.edu (R.K.); 3Institute of Physics, Academia Sinica, Nangang, Taipei 11529, Taiwan; yychen@phys.sinica.edu.tw (Y.-Y.C.); phhwu@gate.sinica.edu.tw (Y.H.)

**Keywords:** cryo-EM, single particle, direct electron camera, nodavirus, icosahedral particles

## Abstract

During this global pandemic, cryo-EM has made a great impact on the structure determination of COVID-19 proteins. However, nearly all high-resolution results are based on data acquired on state-of-the-art microscopes where their availability is restricted to a number of centers across the globe with the studies on infectious viruses being further regulated or forbidden. One potential remedy is to employ multipurpose microscopes. Here, we investigated the capability of 200 kV multipurpose microscopes equipped with a direct electron camera in determining the structures of infectious particles. We used 30 nm particles of the grouper nerve necrosis virus as a test sample and obtained the cryo-EM structure with a resolution as high as ∼2.7 Å from a setting that used electron counting. For comparison, we tested a high-end cryo-EM (Talos Arctica) using a similar virus (*Macrobrachium rosenbergii* nodavirus) to obtain virtually the same resolution. Those results revealed that the resolution is ultimately limited by the depth of field. Our work updates the density maps of these viruses at the sub-3Å level to allow for building accurate atomic models from de novo to provide structural insights into the assembly of the capsids. Importantly, this study demonstrated that multipurpose TEMs are capable of the high-resolution cryo-EM structure determination of infectious particles and is thus germane to the research on pandemics.

## 1. Introduction

The studies of virus structures have played important roles in the historical development of cryo-electron microscopy (cryo-EM). In late 1960, Klug and his colleagues, mainly DeRosier and Crowther, at the Medical Research Council had established the mathematical principles for the reconstruction of three-dimensional structures from two-dimensional electron images using viruses with icosahedral symmetry [1,2]. A much more difficult route for non-periodical objects was pioneered by Hoppe [3] and eventually fulfilled by Frank in the establishment of the single-particle reconstruction as a general method to include proteins with low or no symmetry [4], paving the way for the cryo-EM resolution breakthrough. In mid-1980, Dubochet and his colleagues at the European Molecular Biology Laboratory introduced a plunge-freezing method [5] to preserve biological macromolecules in vitreous ice; by using particles of an icosahedral virus as a cryo-EM specimen, Vogel et al. obtained a structure with a resolution of 35 Å [6], demonstrating that the structure of a biological macromolecule can be rendered “intact” with rapid freezing. In later years, several groups had endeavored to gradually extend the resolutions of cryo-EM for structure determination using icosahedral viruses [7,8,9], until the resolving of the main chain densities in the capsid protein using a high-end 300 kV cryo-EM with the images recorded on films [10]. Due to the ease of sample loading and screening, together with the high stability in electron beam quality and an excellent vacuum to support multiple days of data collection nonstop, this model soon proliferated in the wealthy regions of the world.

Since the resolution revolution of cryo-EM [11], nearly all high-resolution cryo-EM reconstructions (3.5 Å or better), including many virus capsid proteins reported to date, have been obtained using cryo-EMs operated at an accelerating voltage of 300 kV [12,13,14,15,16,17]. Those impressive results are partly associated with a number of advantages provided by transmission electron microscopes (TEMs) with higher accelerating voltages. Those advantages include a greater depth of field (less curvature for the Ewald sphere) [18], a larger mean-free path for electrons, smaller phase errors induced by beam-tilt [19], and a minimized specimen charging effect [20]. However, the benefits provided by high-end 300 kV instruments are not without costs; they are not only expensive to purchase but also costly to maintain, thus often inhibiting many institutions [21]. Consequentially, there is a growing interest in using cryo-EMs with lower accelerating voltages that are more affordable. The recent work by Herzik and Lander [22] on breaking the resolution boundary of 2 Å for the apo-ferritin structure using a 200 kV instrument (Talos Arctica) [23] has again demonstrated the power of a 200 kV instrument. Notably, using a newer 200 kV cryo-EM model with a superior field-emission source, the Subramaniam group obtained a 1.8 Å structure for beta-galactosidase [24]. As the latter specimen is a more challenging sample than apo-ferritin (a conformationally rigid molecule) due to its lower symmetry and conformational flexibility, this work demonstrated that a state-of-the-art 200 kV instrument can rival existing high-end 300 kV models.

Importantly, those spectacular results were all from high-end automated instruments, operated at 200 or 300 kV, and had helped establish and reinforce the view that the instrument performance of an automated cryo-EM is more superior to those established multipurpose TEMs [25] with complete oblivion that those models are known for their high-performance, to support cutting-edge microscopy development [26,27]. As these automated instruments are much more expensive, they are mostly available as shared facilities in a limited number of centers where the access is extremely competitive. In the era of the pandemic, the demand for solving the structures of virus proteins using cryo-EM is surging [28], while the current dilemma is that operating those projects at many centers, including ours, has either been hindered or declined, due in part to the crowding of projects, and in part to that most centers are not designed to accommodate the high level of biosafety required for those infectious materials.

Could there be a remedy? Recently, the capability of 200 kV multipurpose TEMs with a side-entry cryo-holder and a direct electron camera for high-resolution single-particle analysis was demonstrated by Kayama et al. [29], showing that sub-3 Å can be achieved from these established instruments with the usage of apo-ferritin as the test specimen [23]. This work inspired us to investigate the usage of the more affordable multipurpose models for structural studies on infectious particles. It should be pointed out that it has remained largely unknown whether or not the resolution achieved for apo-ferritin (ca. 12 nm) on a 200 kV multipurpose instrument can be similarly obtained for much larger particles. To evaluate the potential of a multipurpose TEM on virus structure determination, we embarked on 30 nm particles of a betanodavirus that could infect a Grouper fish [30]. These fish nodaviruses (NVs) are also called Nerve Necrosis Viruses (NNVs) as they cause an acute syndrome of viral nervous necrosis (VNN), responsible for great mortality at the larval stage among diverse fish species. The Grouper NNV (GNNV) majorly limits the expansion of the Grouper fish culture in Taiwan and worldwide [31,32]. The first high-resolution structure of 30 nm GNNV particles with a triangular number of 3 (T = 3) was made available from X-ray crystals that diffracted to 3.6 Å [33]. Unfortunately, this resolution falls short of the crystallographic resolution. Therefore, one purpose of the present study was to test the capability of cryo-EM with the hope of generating a density map of improved quality so as to support the building of a more accurate atomic model.

Here, we tested two different settings of multipurpose cryo-EM, both with a direct electron camera, for acquiring movie data of frozen-hydrated Dragon GNNV (DGNNV) particles (T = 3, ca. 30 nm in diameter) [30,33]. With global motion correction [34] and careful dose fractionation analysis, followed by classical reconstruction algorithms optimized for icosahedral particles [35,36], the resolutions of the DGNNV from the two settings initially reached 3.5 and 3.1 Å, respectively. We later furthered the resolutions to 3.1 and 2.7 Å, respectively, by the usage of modern algorithms, including patch-wise motion correction [37] and a combination of GPU-enabled refinement methods, to perform dose weighting and cope with high-order aberrations and the contrast transfer function of the individual particle [38,39]. These new results provided the GNNV structures with the highest resolutions to confer the structural information needed for understanding the particle assembly. As we were curious about the performance of a 200 kV automated cryo-EM (Talos Arctica) installed in our newly established cryo-EM center, we tested this cryo-EM with 30 nm particles of a shrimp nodavirus (MrNV) (T = 3), evolutionarily related to GNNV [40]. Remarkably, we also obtained the cryo-EM structure with a resolution of 2.7 Å, updating the 7 Å cryo-EM structure obtained earlier using a 200 kV multipurpose instrument [41] similar to one of our settings. Importantly, this investigation on MrNV allowed us to compare the stage drift of the multipurpose TEMs against that of a high-end cryo-EM. To our surprise, the drift characteristics were comparable. It should be noted that the resolution of 2.7 Å obtained for both the DGNNV and MrNV likely represents the resolution ceiling set by the depth of field conferred by 200 kV electrons. Taken together, our findings update the structures of GNNV and MrNV to sub-3 Å, demonstrating that the performance of a multipurpose TEM with a direct electron camera can rival a high-end cryo-EM and suggesting an affordable alternative for the structural studies of infectious particles.

## 2. Results

### 2.1. Cryo-EM Imaging of DGNNV with Setting A

At the dawn of the cryo-EM revolution [11], we installed a direct electron camera (DE-20, Direct Electron, LP, San Diego, CA, USA)) on a multipurpose TEM (2100 F, JEOL Ltd., Akishima, Tokyo, Japan) with non-negligible spherical aberration (Cs 3.3 mm) suited for developing phase-contrast imaging [42]. We refer to this setting as setting A, from which we obtained a cryo-EM structure of DGNNV at sub-4 Å. This structure was reported in regional conferences in the period of 2014–2015 but was soon shadowed by the X-ray structure (3.6 Å) [33] and the cryo-EM structure (3.9 Å) [43], obtained by a 300 kV high-end cryo-EM with a CCD that precluded image de-blurring by motion correction [34]. Nonetheless, our early efforts on processing the movie data from a direct electron camera had conferred valuable insights into how to push resolutions by mitigating the effects of stage and specimen motion and radiation damage by 200 kV electrons.

With this setting A that used a side-entry holder (Gatan 626, Gatan Inc., Pleasanton, CA, USA), we performed cryo-EM imaging of DGNNV particles using the linear mode provided by the direct electron camera, which has 5120 by 3840 sensors of 6.4 microns in size where, at the nominal microscope magnification of 50,000×, a pixel corresponded to 1.16 Å (Table 1 and Appendix A). Each micrograph was composed of 50 frames in 2 s with the dose rate set to 20 e^−^/Å^2^ per second over the specimen. For individual micrographs, the drift for each frame was analyzed globally and corrected by using the DE_process_frame.py software from the vendor (Direct Electron LP, San Diego, CA, USA). As shown in Figure 1A, the particles in the “motion-corrected” micrograph became more visible and the contrast transfer function in the rotationally averaged power spectra was extended (Appendix A). The image drift was in the range of an overall translation of 10–20 Å over 2.0 s in an indefinite direction (Appendix A), reflecting the instability typical of the side-entry holder. Larger steps of drift were observed in initial frames (Appendix A) as the initial movement was predominately beam-induced [44].

Figure 1B and Appendix A present the cryo-EM structure of DGNNV obtained from the icosahedral reconstruction from 41,318 particle images in motion-corrected micrographs resulting from 1.5 s movies. Initially, even with motion correction (Appendix A), the overall resolution was stalled at 4.21 Å with a total accumulated dose of ~30 e^−^/Å^2^. To improve the resolution, we investigated the radiation damage effect by dose fractionation analysis using frame exclusion [34]. We first split the 24 frames in the first second (the first frame excluded) into two consecutive 12-frame segments to obtain the early-time and late-time reconstructions. As expected, the overall resolution for the late-time reconstruction deteriorated (Appendix A), consistent with the diminished side-chain density of arginine in a helix (Appendix A). This result suggested significant damage had occurred prior to 20 e^−^/Å^2^. To pinpoint the critical dose, we made a finer fractionation. As shown in Figure 1C, the best overall resolution of 3.79 Å was achieved by limiting the total dose to ~14 e^−^/Å^2^. Figure 1D displays the maps of helices, sheets, and a loop of the capsid protein, supporting the reported resolution. To further validate the map, we replaced the His94 with Ala to obtain a mutant reconstruction, in which the bulky side chain of His94 is missing (Figure 1E), confirming that the density at the corresponding position in the wild-type reconstruction is indeed attributed to His94.

The above structural studies of the DGNNV from setting A were carried out using CPU-based algorithms optimized for icosahedral particles [35,36] with the resolution only arriving at ~3.5 Å, corresponding to two-thirds of the Nyquist frequency. This implies there is still room for improvement. To this end, we employed advanced methods [38] that can eliminate residual phase errors or aberrations. We then re-processed the data from setting A with a combination of RELION [38] and cryoSPARC [39] with the newest version released recently. These algorithms are empowered by graphical computation units (GPUs), while the packages are being continually updated. With the advent of high-quality GPU cards offering good size memory, it is possible to accommodate the images of DGNNV particles. Importantly, these algorithms provide dose weighting schemes to compensate for the radiation damage so that there is no need to perform the dose fractionation with frame exclusion as described earlier. In addition, these algorithms can couple with the patch-wise motion correction [37] to generate better motion-correction results than the global correction [34]. As summarized in Table 1, reprocessing the data of setting A has indeed extended the resolutions: for the wild-type and H94A mutant DGNNV particles with the resolutions reaching 3.24 and 3.15 Å, respectively (also see Appendix A), surpassing that from the X-ray crystal (3.6 Å) [33].

### 2.2. Cryo-EM Imaging of DGNNV with Setting B

After we obtained the DGNNV structures from setting A in 2014, another direct camera named K2 (Gatan Inc., Pleasanton, CA, USA) started to dominate the community due to its superior quantum efficiency when operated in counting mode [26,34]. However, K2 mostly serves as an asset to a high-end instrument except that at a few places including Stanford it is equipped on a multipurpose TEM. This unique setting at Stanford (setting B) combines a 200 kV EM (Technai F20, FEI, Hillsboro, OR, USA) with a side-entry holder (Gatan 626, Gatan Inc., Pleasanton, CA, USA) and a K2 direct electron camera without an energy filter (Gatan Inc., Pleasanton, CA, USA). To assess the performance of K2, we used setting B to image DGNNV particles in 2016. To increase data collection efficiency by maximizing the number of particles per micrograph, we used a low magnification (29,000×) on F20 while operating K2 at the super-resolution mode with the physical resolution set at 1.24 Å/pixel (Table 1 and Appendix A). The imaging parameters were set similar to setting A except that a small spot size (no. 6) was used to provide reduced flux for minimizing coincidence loss [34]. The data were processed in the same manner as for setting A to ensure a fair comparison. Nonetheless, the resulting coherence of setting B seemed to be better than setting A. By first binning the data by 2×, we obtained the cryo-EM structure of DGNNV with an overall resolution of ~3 Å (Appendix A). With the orientation parameters obtained from the binned data, we refined the un-binned (super-resolution) data slightly beyond 3 Å (2.86 Å) (Appendix A).

Figure 2 highlights some key features of the cryo-EM structure of DGNNV from setting B. In Figure 2A, we deliberately removed the protrusion domains to illustrate the fineness of the shell of the particle. Compared to the map of DGNNV from setting A (Figure 1D), the map quality from setting B is evidently improved (Figure 2B,C), consistent with the reported resolutions (Figure 2D,E). We recently further re-processed the data from setting B with RELION [38] and cryoSPARC [39]. Interestingly, the resolution of DGNNV using the 2× binned data was extended to 2.72 Å as summarized in Table 1 and Appendix A. Our attempts to re-process the super-resolution data have encountered the challenge associated with insufficient memory since each un-binned particle image occupies approximately 600 by 600 pixels, several times larger than a medium-size protein complex that suits the GPU hardware.

### 2.3. Comparision of DGNNV Maps from the Present and Earlier Studies

To compare our NNV cryo-EM structure with the existing structures, we display them side-by-side in Figure 3, in which the left, middle, and right column represent our cryo-EM structure of DGNNV (<3.0 Å), the earlier cryo-EM structure of GNNV from Orange-spotted Grouper (3.9 Å) [43], and the X-ray structure of GNNV from Red-spotted grouper (3.6 Å) [33], respectively. As shown in the first row in Figure 3, the overall size of GNNV in the X-ray structure (Figure 3C) is smaller than those in the two cryo-EM structures due to the protrusions assuming a more compact conformation. Currently, the origin and the function of this compactness are unknown. As shown in the second (Figure 3D–F) and the third row (Figure 3G–I), the map of the shell region obtained by using a multipurpose cryo-EM with a direct electron camera turns out to have the highest quality (Figure 3D,G). The improvement of our cryo-EM structure of GNNV (Figure 3A,D,G) from an earlier cryo-EM structure from CCD imaging (Figure 3B,E,H) [43] is evident. The improvement is attributed to the direct electron technology.

A previous functional study on GNNV has suggested that the NTD (N-terminal domain) could stabilize the icosahedral VLPs mainly by the intra-molecular and inter-molecular hydrogen bonding [30]. The corresponding densities of the X-ray structure of RGNNV were too obscure to support the modeling of the bonding [33]. To assess whether this obstacle might be mitigated with the advancement of resolution, we compared the maps for this region. As shown in Figure 4, only part of the NTD (residues 35–51) could be modeled in the present and earlier structures [33,43]. Nonetheless, as we compared the density map (Figure 4), we found that our map (Figure 4A) offered the best-resolved density as opposed to the other two maps (Figure 4B,C). For example, we observe the side chain orientation of Phe48 in our cryo-EM model is different from that in the X-ray model. In the X-ray structure, the electron density of the Phe48 residue displays a round shape (Figure 4B) such that the rotamer cannot be uniquely defined. By contrast, the density of Phe48 in our cryo-EM map exhibits a flat shape such that there is little ambiguity in defining the rotamer (Figure 4A). This finding indicates that extending the resolution to 3 Å or better is crucial for building an accurate atomic model.

### 2.4. The Usage of the Automated Instrument Only Improves Efficiency

We installed high-end cryo-EMs at Academia Sinica in 2018. During the installation period, we chose to test Talos Arctica (200 kV) with a virus-like particle we derived from *Macrobrachium rosenbergii* nodavirus (MrNV). This MrNV virus is closely related to NNV [40]; it causes White tail disease (WTD) by affecting shrimp larvae at post-larvae and early juvenile stages, causing up to 100% mortalities. Recently, the structure of T = 3 shrimp nodaviruses has been determined by a multipurpose 200 kV cryo-EM with a DE-20 camera to 7 Å [41].

Figure 5 summarizes the cryo-EM structure of MrNV obtained from our high-end cryo-EM with a direct electron camera (Falcon III, Thermo Fisher Scientific, Hillsboro, OR, USA) operated in linear mode. It was noted that the apparent coherence from this setting (Talos-3EC) seemed to be poorer (Appendix A) than setting A and B. The overall resolution reached 2.92 Å (Appendix A), by which the atomic model could be built from de novo without the need of homologous modeling as resorted to in an earlier study [40]. This resolution corresponds to 0.58 Nyquist as these data were collected with a fine pixel size (0.86 Å) by using a high magnification. Figure 5G highlights the well-resolved densities of the calcium ions, consistent with the previous finding [40]. We then further re-processed these data using the newest version of RELION [38] and CryoSparc [39] as we did for the data of settings A and B to extend the resolution to 2.70 Å (Table 1 and Appendix A). It should be noted this milestone of resolution achieved for MrNV is identical to the highest resolution obtained for GNNV. 

## 3. Discussion

In this study, we have explored the potential of using a multipurpose 200 kV field-emission TEM with a direct electron camera for high-resolution studies of large virus particles. The tested electron microscopes are from two different vendors (JEOL and FEI) but have comparable instrument performance in terms of electron-optics and stage stability. They were equipped with a direct electron camera from different vendors (DE-20, Direct Electron LP, San Diego, CA, USA; K2, Gatan Inc., Pleasanton, CA, USA) to enable the images to be recorded as movies, wherein the second setting of the K2 camera was operated in the counting mode [34]. The earlier processing of the movie data using the then-advanced motion correction and dose fractionation analysis [34] has largely mitigated the adverse effects of stage drift and radiation damage, yielding cryo-EM structures of DGNNV at 3.5 Å and 3.1 Å from these two settings, respectively. For the second setting, a slightly better result (2.86 Å) was obtained by using super-resolution data. With the recent advanced GPU-enabled algorithms [38,39], the resolutions from the two settings were further extended to 3.1 Å and 2.7 Å, respectively. In both cases, the resolutions surpass that of the X-ray diffraction [33]. Our attempts of furthering the resolution by processing the super-resolution data using GPU-enabled algorithms [38,39] have not been met with success, due to the issue associated with very large image size. Nonetheless, we expected that the improvement, if any, would be marginal.

Our tests on the two settings of multipurpose TEMs showed that the results from setting B with electron counting were better than setting A by 0.4 Å, which is consistent with the observations made by Kayama et al. in comparing K2 and DE-20 [29]. However, we cannot entirely attribute the higher resolution to the superior DQE of the counting camera [26], as higher coherence was achieved on this setting (F20-K2) also (Appendix A) by using a fairly small spot size on the source, by which the beam flux was reduced for minimizing the coincidence loss [34]. By contrast, the coherence of the other setting (2100F-DE20) (Appendix A) was compromised by the usage of a larger spot, circumstantially selected in exchange for the increase in beam flux from the aging field-emission source. Besides, the spherical aberration on 2100 F (Cs 3.3 mm) was slightly poorer than that of F20 (Cs 2.3 nm) as the former was intended for the use of a phase plate [42] similarly encountered on a setting in the study by Kayama et al. [29].

A question arises as to how far one can push on these multipurpose TEMs. Using a high-end cryo-EM of the same accelerating voltage (Talos Arctica) on a similar particle (MrNV), the same final resolution of 2.7 Å was obtained, suggesting 2.7 Å would represent a fundamental limit for 30 nm particles ultimately attainable from a 200 kV instrument. What would then be the limiting factors? As pointed out in a seminal paper by Zhang and Zhou [45] catalyzing the cryo-EM revolution, the resolution of a cryo-EM structure is a cumulative outcome from a number of limiting factors. Those factors include those germane to the specimen and others to the instrument, including stage stability and aberrations caused by imperfections of beam alignment [19]. Our studies showed that the drifts of the stage and specimen in combination on the multipurpose TEMs were comparable to those on Talos Arctica, and that the resulting adverse effect could be largely eliminated by applying motion correction to the movie data. To minimize the high-resolution phase errors induced by beam tilt, the multipurpose TEMs were both well-aligned for achieving parallel beam illumination [19]. However, the elimination of coma by additional coma-free alignment [19] was not performed on those multipurpose TEMs, but on the high-end 200 kV cryo-EM (Talos Arctica) with the usage of the software from the vendor (Thermo Fisher Scientific, Hillsboro, OR, USA). Nonetheless, the residual phase errors resulting from imperfect alignment, if any, were likely to have been largely corrected “in silico” [22] by the advanced software used in this study [38]. By ruling out those mentioned factors, the remaining factor was the focus gradient across the height of the specimen. Along this line, we found that the figure of 2.7 Å coincided with the theoretical limit set for a 30 nm particle with 200 kV electrons (see Table 4 in [45] and the reference therein). It is possible to use Ewald sphere correction [18,45,46] to lift this resolution ceiling, which has been demonstrated by a sub-2Å structure of adenovirus using a 300 kV instrument [47]. Nonetheless, there is a case of a better-than-3.5Å resolution obtained for a 75 nm icosahedral particle without using this correction [48].

Recently, there has been increasing interest in using cryo-EM of lower accelerating voltages where the results from 200 kV cryo-EMs of high-end models start to rival those from 300 kV instruments [22,24]. It should be noted that the 1.7 Å was obtained for apo-ferritin by using Talos Arctica [22]. Interestingly, this figure perhaps reflects the resolution ceiling also imposed by the focus gradient on a 12 nm particle as it matches exactly the value calculated using the same formula used for generating the numbers in Table 4 in [45]. Were Ewald sphere correction to become a routine [46], one may expect similar results would even be attainable from an even more affordable 100 kV instrument [49] given that a direct electron camera optimized for 100 kV electrons is available.

In summary, we demonstrate that the performance of multipurpose TEMs is comparable to a high-end cryo-EM. In addition, we provide sub-3 Å structures for GNNV and MrNV for shedding light into their particle assembly. Concerning that a high-end cryo-EM has the advantage in data throughput via automation, it is now possible to achieve similar efficiencies on multipurpose TEMs by using open-source hardware-controlling software such as Leginon [50] or those related to Serial EM [51,52] for automated data collection. Besides, it is pertinent to use a new cryo-holder on multipurpose TEMs that can keep the specimen at low temperature overnight without human attention. Those features for remedy have been either demonstrated or discussed in the work by Kayama et al. [29]. With the access to cryo-EM centers being restricted, our work demonstrating that multipurpose TEMs combined with a direct electron camera is competitive suggests multipurpose TEMs offer an alternative choice. As multipurpose TEMs are much more affordable, the purchase and installation would meet much fewer financial obstacles. As a result, having a dedicated instrument of multipurpose TEM in a laboratory of high-level biosafety would be appealing. It may seem odd to equip a multipurpose TEM with a high-end camera that is as expensive as the TEM. Fortunately, the high-end cameras are now available in the refurbished form at a significantly reduced price.

## 4. Materials and Methods

### 4.1. Protein Preparation of DGNNV

DGNNV VLPs were expressed as previously described [30] with some modification. The *E. coli* BL21 containing the DGNNV wild-type or mutated capsid protein gene was grown in 2.5 L of LB broth (contain 100 μg/mL of ampicillin) at 30 °C. When the cell density reached ~0.6 OD600 nm, 1 mM of IPTG (Merck-Sigma-Aldrich, Darmstadt, Germany) was added for induction and the incubation temperature was adjusted to 20 °C. After overnight (~16 h) induction, the *E. coli* BL21 cells were harvested by centrifugation at 4500 rpm for 1 h at 4 °C (Avanti J-26XP centrifuge with rotor: JLA8.1, Beckman Coulter, Indianapolis, IN, USA,), and the cell pellets were re-suspended in 100 mL of TN buffer (50 mM of NaCl, 50 mM of Tris-HCl, pH 8.0). The re-suspended cells (IPTG induced) were incubated with 0.5% Triton X-100 (Merck-Sigma-Aldrich, Darmstadt, Germany) at 4 °C for 1 h and lysed by flushing three passages through a French press (Avestin Emulsiflex-C5, ATA Scientific Ltd, Taren Point, NSW, Australia). The insoluble fraction was eliminated by centrifugation at 20,000 rpm for 1 h at 4 °C (Avanti J-26XP centrifuge with rotor: JA-25.50, Beckman Coulter, Indianapolis, IN, USA,). The collected supernatant was centrifuged at 30,000 rpm for 3.5 h at 4 °C (OptimaTM L-90K Ultracentrifuge with rotor: SW 41 Ti, Beckman Coulter, Indianapolis, IN, USA,) against a 30% (*w*/*w*) sucrose cushion. After the ultra-centrifugation step, the supernatant was removed and the VLP pellet was re-suspended in 200 μL of TN buffer. The suspended pellet was layered onto a 10 mL 10–40% (*w*/*w*) sucrose density gradient pre-generated by a gradient maker (Gradient Master, Biocomp, Fredericton, NB, Canada) and centrifuged at 30,000 rpm for 3.5 h at 4 °C (OptimaTM L-90K Ultracentrifuge with rotor: SW 41 Ti, Beckman Coulter, Indianapolis, IN, USA,). The VLP-containing fractions (500 μL) were collected and diluted with TN buffer (pH 8.0). An Amicon Ultra-0.5 (100 kDa, Merck-Millipore, Darmstadt, Germany) was used to concentrate the sample, and the resultant purified VLPs were analyzed by SDS-PAGE and negative-stained electron microscopy. The concentration of the VLPs was adjusted to ~100 μg/mL for cryo-EM imaging.

### 4.2. Cryo-Electron Microscopy of DGNNV

To prepare a cryo-EM sample, an aliquot (~4 μL) of protein was placed onto a Quantifoil R1.2/1.3 holey carbon grid (Quantifoil Micro Tools GmbH, Jena, Germany) with a layer of thin carbon film, manually blotted with filter paper for 1.2 s (Whatman No.1, Merck-Sigma-Aldrich, Darmstadt, Germany), and plunged into liquid nitrogen-cooled liquid ethane. All subsequent steps were carried out below −170 °C to prevent de-vitrification. Grids were transferred to a Gatan 914 or 626 side-entry cryo-holder (Gatan Inc., Pleasanton, CA, USA), and examined in a 200 kV JEM-2100F electron microscope (JEOL Ltd., Akishima, Tokyo, Japan) equipped with a DE-20 camera (Direct Electron, LP, San Diego, CA, USA). The camera was mounted in the film port on our electron microscope operated at 200 kV. Cryo-EM images were recorded with a magnification of 50,000× in “movie mode” with an exposure time of 2.0 s that contained 50 frames. The corresponding pixel was 1.16 Å and the dose rate was ~20 e^−^/Å^2^ per second to yield a total accumulated dose of ~40 e^−^/Å^2^ (~0.8 e^−^/Å^2^ per frame). The defocuses of cryo-images were set in the range from ~0.5 to ~3 µm.

### 4.3. Data Processing of DGNNV

The determination of each frame displacement and subsequent motion correction by frame alignment was performed using the program DE_processed_frame.py distributed with the DE-20 camera (Direct Electron, LP, San Diego, CA, USA). A total of 41,318 VLPs particles from 588 digital micrographs were semi-automatically picked using the e2boxer.py program in the EMAN2 suite [36]. After obtaining the particle coordinates, the particles were extracted from the cryo-EM micrographs and the boxed particles were normalized by the Robem program distributed with Auto3dem [53]. The values of defocus and astigmatism were estimated by CTFFIND3 [54]. The initial model was generated by the “ab initio random model” method [53], in which a random orientation was assigned to each particle in the raw data set. In the step of orientation search, refinement was performed by the PPFT and PO2R program in the Auto3dem package with a random initial model, as described above, and the 3D icosahedral symmetric reconstruction was performed by the P3DR program with the “symm_code 532” flag [53].

In order to obtain a realistic resolution estimation for cryo-EM structure determination, the gold-standard Fourier shell correlation (FSC) with the 0.143 cutoff criterion [55] was used. The whole DGNNV dataset was first divided into two halves and processed independently for all the subsequent reconstruction procedures. In each dataset, the initial models were reconstructed by the “ab initio random model” method [53] in which random orientations were assigned to the particles to generate the first initial model. The initial models in each separate data set were subsequently refined iteratively until no more improvement in the resolution estimated from those two independent reconstructions. The final 3D map was reconstructed from the whole dataset.

### 4.4. Map Sharpening

Before map segmentation and atomic model building, the density maps of whole virus particles were subjected to map sharpening according to the approach introduced by Rosenthal and Henderson [56]. The B-factor estimation and restoration of the decay of the amplitudes were performed in the EM-BFACTOR program [57] by applying the inverse of the B-factor to sharpen the map. After EM-BFACTOR sharpening, both the densities of the main chain and side chain were better resolved and the sharpened density maps of whole virus particles were subjected to map segmentation and atomic model building.

### 4.5. Map Segmentation and Model Building

The reconstructed 3D cryo-EM map was visualized in UCSF Chimera [58] and three quasi-equivalent capsid protein subunits (A, B, and C) within a T = 3 icosahedral lattice were segmented by Chimera based on the density connectivity. In order to reduce noise and enhance the connectivity of the peptide chain within a subunit, the three segmented subunits were aligned and averaged. The alignment was performed using the “fit in map” utility in Chimera, which performed the translation and rotation of the density map. The aligned segmented subunits (A, B, and C) were then averaged to obtain an averaged subunit map, based on which the peptide chain tracing was performed. The initial atomic model of the capsid protein was de novo built in the COOT package [59] with “Add Terminal Residue”, “Simple Mutate”, and “Real Space Refinement Zone” functions in the “Model/Fit/Refine” utility. The residue assignment was guided by bulky residues such as histidine, tyrosine, phenylalanine, and tryptophan. The unique patterns of residues in amino acid sequences were also taken into consideration for model building and validation. After manually building chain A, the conserved atomic model built from the averaged subunit map, the atomic model of chain A was copied and fit into the density of each subunit (A, B, and C) within an icosahedral asymmetric unit (ASU) using the “fit in map” utility in Chimera. The atomic models for each subunit (A, B, and C) were examined in the COOT program for visual identification of the side-chain densities. The side chains in the atomic models that did not match the cryo-EM density were adjusted in the Coot program manually by the “Real Space Refinement Zone” function with the restraints of “Torsion”, “Plannar Peptide”, “Trans Peptide”, and “Ramachandran”. The atomic model of an asymmetric unit (ASU) was then further refined by PHENIX [60] using the “Real-space refinement” function with the default setting with the input atomic model, cryo-EM map, and resolution valve estimated by gold-standard FSC. The PHENIX refined atomic model was subsequently visually inspected in COOT, and the problematic regions and Ramachandran outliers were manually corrected by “Real Space Refinement Zone”, as described above. Several runs of “Real-space refinement” of the atomic model were performed in COOT and PHENIX until no further improvement. The residues missing in the N-terminal domain (NTD) and in the low-resolution protrusion domain were not modeled. The validation of the atomic model was performed in PHENIX with the “Comprehensive validation (cryo-EM)” function. The validation statics are summarized in Appendix A.

### 4.6. Protein Preparation of MrNV

The capsid protein of MrNV Taiwanese strains (residue 1–371, Gene Bank accession number: ABG25924) was obtained through full gene synthesis and cloned into *E. coli* BL21 with primers for over-expression. Briefly, the *E. coli* containing the MrNV capsid protein gene was grown in 2.5 L of LB broth (contain 100 μg/mL of ampicillin) at 30 °C. When the cell density reached ~0.6 OD600 nm, 1 mM of IPTG (Merck-Sigma-Aldrich, Darmstadt, Germany) was added for induction and the incubation temperature was adjusted to 20 °C. After overnight (~16 h) induction, the *E. coli* cells were harvested by centrifugation at 4500 rpm for 1 h at 4 °C (Beckman Coulter, Indianapolis, IN, USA, Avanti J-26XP centrifuge, rotor: JLA8.1) and the cell pellets were re-suspended in 100 mL of TN buffer (50 mM of NaCl, 50 mM of Tris-HCl, pH 8.0). The re-suspended cells (IPTG induced) were incubated with 0.5% Triton X-100 (Merck-Sigma-Aldrich, Darmstadt, Germany) at 4 °C for 1 h and lysed by flushing three passages through the French press (Avestin Emulsiflex-C5, ATA Scientific Ltd, Taren Point, NSW, Australia). The insoluble fraction was eliminated by centrifugation at 20,000 rpm for 1 h at 4 °C (Beckman Coulter, Indianapolis, IN, USA, Avanti J-26XP centrifuge, rotor: JA-25.50). The collected supernatant was centrifuged at 30,000 rpm for 3.5 h at 4 °C (Beckman Coulter, Indianapolis, IN, USA, Optima L-90K Ultracentrifuge, rotor: SW 41 Ti) against a 30% (*w*/*w*) sucrose cushion. After the ultra-centrifugation step, the supernatant was removed and the pellet was re-suspended in 400 μL of TN buffer. The suspended pellet was layered onto a 10 mL 10–40% (*w*/*w*) sucrose density gradient pre-generated by a gradient maker (Gradient Master, Biocomp, Fredericton, NB, Canada) and centrifuged at 30,000 rpm for 3.5 h at 4 °C (Beckman Coulter, Indianapolis, IN, USA, Optima L-90K Ultracentrifuge, rotor: SW 41 Ti). The VLP-containing fractions (500 μL) were collected and diluted with TN buffer (pH 8.0). An Amicon Ultra-0.5 (100 kDa, Merck-Millipore, Darmstadt, Germany) was used to concentrate the sample, and the resultant purified VLPs were analyzed by SDS-PAGE and negative-stained electron microscopy. The concentration of the MrNV VLPs was adjusted to ~100 ng/μL for cryo-EM imaging.

### 4.7. Cryo-Electron Microscopy of MrNV

To prepare a MrNV cryo-EM grid, an aliquot (~3.5 μL) of protein (~100 ng/uL) was placed onto a Quantifoil R1.2/1.3 holey carbon grid coated with thin carbon film (Quantifoil Micro Tools GmbH, Jena, Germany), blotted with filter paper for 3 s, and plunged into liquid nitrogen-cooled liquid ethane on a Vitrobot station (Mark IV, Thermo Fisher Scientific, Hillsboro, OR, USA). All subsequent steps were carried out below −170 °C to prevent de-vitrification. Grids were clipped, mounted in a magazine, and transferred with a nano-cap to Talos Arctica (Thermo Fisher Scientific, Hillsboro, OR, USA), a 200 kV cryo-EM equipped with a new generation of X-FEG (Thermo Fisher Scientific, Hillsboro, OR, USA). Careful alignment was performed in nanoprobe mode (spot size 3, gun lens 4 with C2 lens set at 43.8%) to achieve parallel beam alignment and coma-free alignment. Cryo-EM images were recorded at a magnification of 120,000× with Falcon III (Thermo Fisher Scientific, Hillsboro, OR, USA) operated in linear mode with a pixel size of 0.86 Å. The defocuses used for imaging were set from ~1.0 to ~2.5 µm. For imaging MrNV, the dose rate was set to 12 e^−^/Å^2^ per second to yield a total dose of 37 e^−^/Å^2^ in three seconds (divided into 30 frames). A total of 1353 movies were collected using EPU (Thermo Fisher Scientific, Hillsboro, OR, USA) over a period of 22 h.

### 4.8. Data Processing of MrNV

Micrograph de-blurring was achieved for each movie using MotionCorr2 [37]. The defocus of the “motion-corrected” micrograph was estimated by Gctf [61]. The automatically picked up particles from the micrographs were screened by 2D classification with the GPU-accelerated RELION 2.0 [62]. A total of 27,905 MrNV particles were selected and then used for ab initio 3D model reconstruction using cryoSPARC [39]. The 3D reconstruction was used as a starting model to initiate the 3D classification on RELION 2.0 to further separate the structural heterogeneity into different 3D classes. The structurally homogeneous particles in a 3D class were used for 3D refinement using 3D auto-refine on RELION 2.0. To obtain realistic resolution estimation for the cryo-EM structure, goldstandard Fourier shell correlation (FSC) with the 0.143 cutoff criterion [55] was used, and the local resolution estimation was performed by ResMap [63].

### 4.9. Data Reprocessing with Modern Algorithms

The image stacks were motion-corrected and dose-weighted using MotionCor2 with a 5 × 5 patch [37]. Contrast transfer function (CTF) information was estimated from the images after motion correction and dose weighting by CTFFind4 [64]. Semi-automated reference-free particle picking was performed by cisTEM [65]. The coordinates of the selected particles were imported into Relion 3.0 [38] for particle extraction. Several runs of 2D classification were performed in Relion 3.0 followed by particle selection and extraction. The ab initio models of VLP were calculated by cisTEM and then imported into Relion for further 3D auto-refinement with icosahedral symmetry. After CTF refinement and Bayesian polishing, the polished shiny particles were imported into cryoSPARC [39] for further 2D classification and 3D homogeneous refinement with icosahedral symmetry. Map sharpening and resolution estimation were also performed in cryoSPARC (Appendix A).

## Figures and Tables

**Figure 1 ijms-22-06859-f001:**
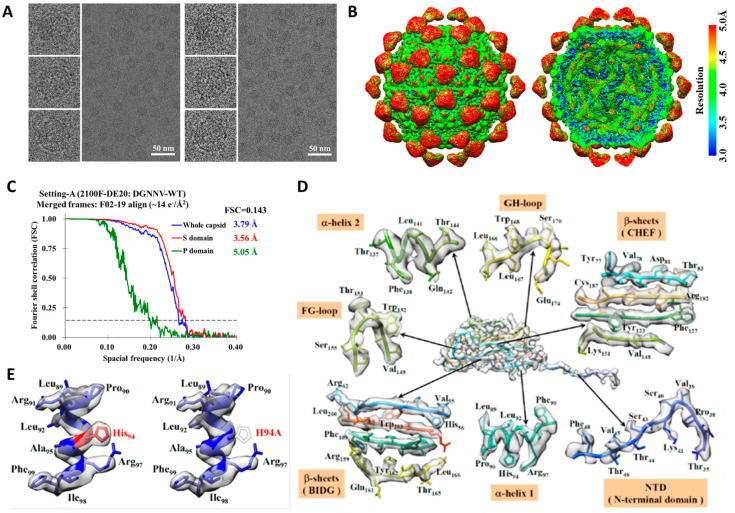
The cryo-EM structure of DGNNV particle from setting A. (**A**) The left is a representative raw micrograph and the right is after whole-micrograph motion correction. (**B**) Overall cryo-EM structure of DGNNV virus-like particle and the cutaway view to visualize the inside. The color spectrum indicates the estimated local resolutions. (**C**) The gold standard FSC curves show the overall resolutions of the whole capsid, the shell (S-domain) and the protrusion (P-domain). (**D**) The density maps of α-helices, β-sheets, and loops extracted from the S-domain of the capsid protein, displayed in the center. (**E**) Comparison of a local density map of wild DGNNV and that of the mutant of H94A.

**Figure 2 ijms-22-06859-f002:**
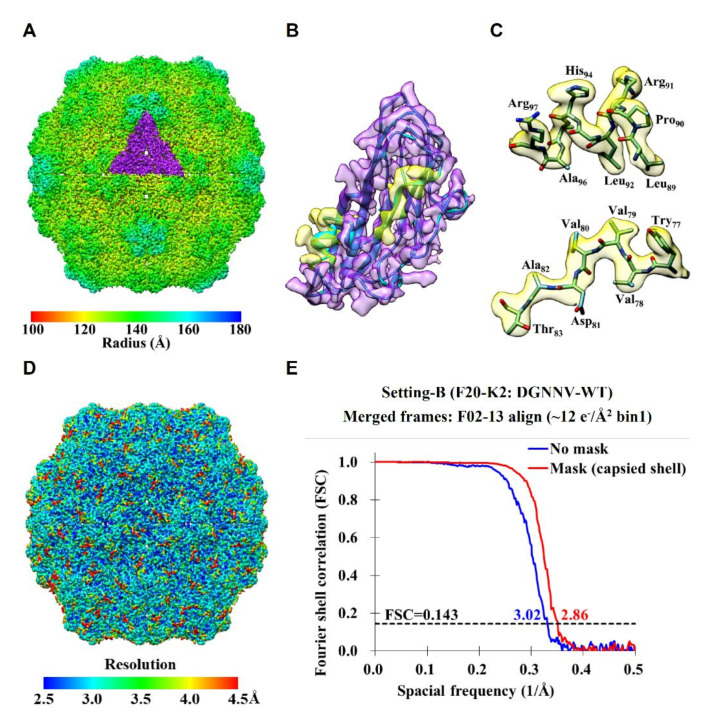
The sub-3Å cryo-EM structure of DGNNV particle from setting B. (**A**) The overall cryo-EM structure of DGNNV with the protrusions being computationally removed. The color spectrum from red to blue indicates the distance from the center where an asymmetric unit (ASU) of three capsid proteins segmented from the particle is colored in purple. (**B**) The density of capsid protein with S-domain only is in purple where the backbone within presented as ribbon is in cyan, and the selected α-helix and β-sheet are in transparent yellow. (**C**) Representative α-helix and β-sheet colored in transparent yellow. The atoms of oxygen, nitrogen and carbon are colored in red, blue and cyan, respectively. (**D**) Local resolution map of the DGNNV reconstruction (see also Appendix A). (**E**) Overall resolutions of DGNNV reconstruction estimated using gold-standard FSC for the un-binned data (bin 1×) processed by the classical algorithms [35,36].

**Figure 3 ijms-22-06859-f003:**
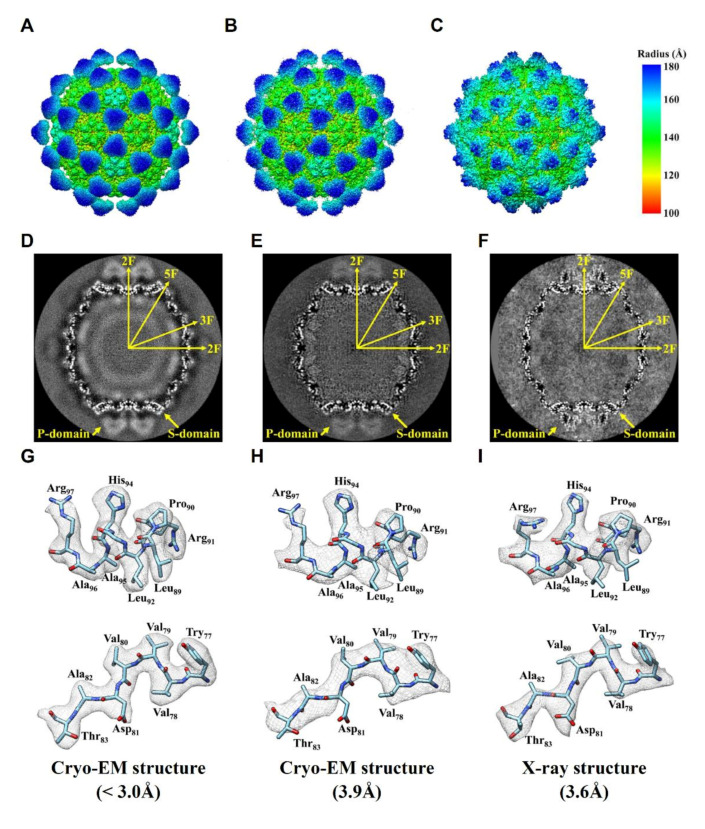
Comparison of the NNV structures. (**A**–**C**) From left to right are our DGNNV cryo-EM structure (from setting B), earlier OSGNNV cryo-EM structure, prepared from EMDB-6453 and PDB:3JBM, and RGNNV X-ray structure, prepared from PDB:4WIZ), respectively. The color spectrum indicates the distance from the center. (**D**–**F**) The corresponding views of central sections. (**G**–**I**) Representative α-helix and β-sheet. The density map is represented in gray mesh. The atoms of oxygen, nitrogen and carbon are colored as red, blue and cyan, respectively.

**Figure 4 ijms-22-06859-f004:**
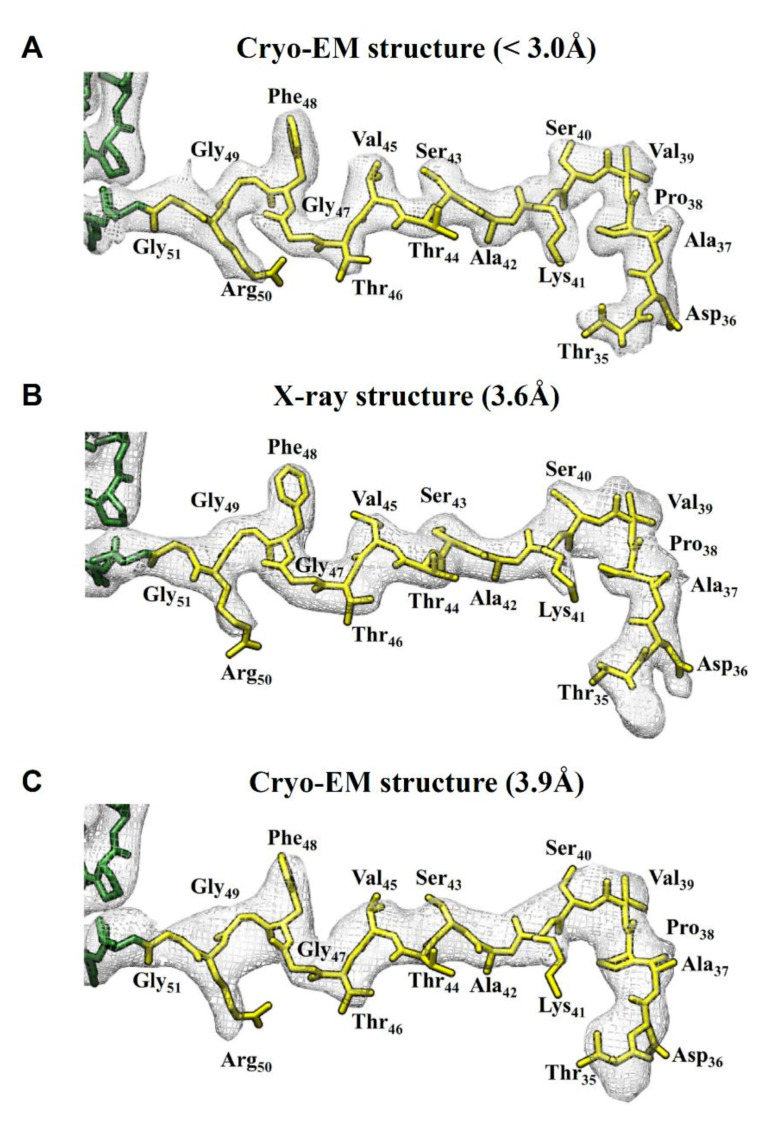
Comparison of the density maps of the N-terminal domain (NTD). From top to bottom are (**A**) the cryo-EM structure of DGNNV (<3.0 Å), (**B**) the X-ray structure of RGNNV (3.6 Å) (prepared from PDB:4WIZ) [33], and (**C**) the cryo-EM structure of OSGNNV (3.9 Å) (prepared from EMDB-6453 and PDB:3JBM) [43]. They are presented from top to bottom in descending order of resolution. The density maps are displayed by gray mesh. The model of the capsid protein in colored in green, while that of the NTD is highlighted in yellow.

**Figure 5 ijms-22-06859-f005:**
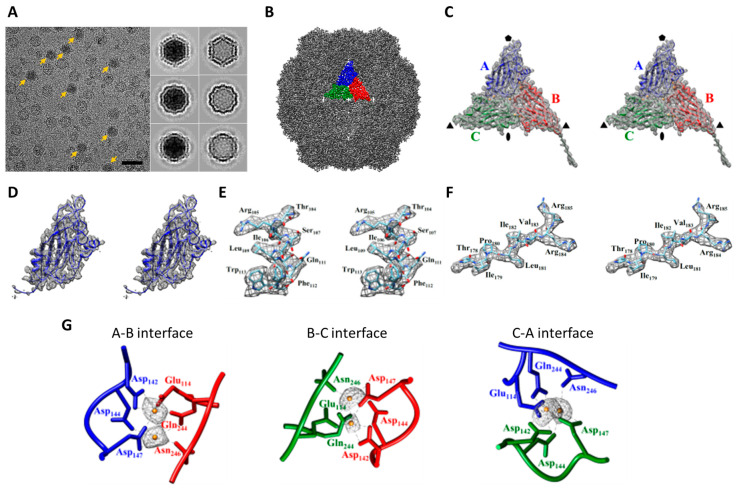
The cryo-EM structure of MrNV particle from Talos Arctica. (**A**) A typical cryo-EM micrograph of ice-embedded MrNV particles. The full MrNV particles are indicated by yellow arrows. The selected 2D class-averaged images of MrNV full and semi-empty particles are shown in the insets. Bar: 50 nm. (**B**) Surface view of MrNV semi-empty particle density map colored in gray. An asymmetric unit of three subunit is highlighted and each subunit is colored differently: subunit A in blue, B in red, and C in green. (**C**) Stereo pair of enlarged views of an asymmetric unit containing subunits A, B, and C with the atomic models enveloped within. (**D**) A stereo pair of the enlarged views of a subunit (subunit A) with the atomic models enveloped within. (**E**) A stereo pair of a α-helix in subunit A composed of residues 104–113; the side chain details are visualized in the segmented density (gray mesh). (**F**) A stereo pair of a β-strand composed of residues 232–237 in subunit A; the side chain details are visualized in the segmented density (gray mesh). (**G**) The models of the side-chains of the calcium-binding motif and other coordinating residues together with the electron density of two putative calcium ions at the A-B, B-C, and C-A interfaces are displayed (subunit A in blue, B in red, and C in green).

**Table 1 ijms-22-06859-t001:** Cryo-EM 3D refinement and statistics.

Data Set	Setting A (2100F-DE20) (DGNNV-WT)		Setting B (F20-K2) (DGNNV-WT)		Setting C (Talos-Falcon3 EC) (MrNV)	
Software	Auto3dem	Relion v3 and cryoSPARC v2	Auto3dem	Relion v3 and cryoSPARC v2	cryoSPARC v1 and Relion v2	Relion v3 and cryoSPARC v2
Selected frames (no.)	18	50	12	50	19	30
Electron exposure (e^−^/Å2)	~14	~40	~12	~50	~23	~37
Dose weighting	No	Yes	No	Yes	No	Yes
Micrographs stacks (no.)	588	588	1009	1009	1353	1353
Final particle images (no.)	41,318	46,796	29,575	41,152	19,049	27,733
Symmetry imposed	I (532)	I	I (532)	I	I (I 1)	I
Pixel size (Å/pixel)	1.16	1.16	1.24	1.24	0.86	0.86
Nyquist limit (Å)	2.32	2.32	2.48	2.48	1.72	1.72
Map resolution of shell (Å) ^1^	3.56	3.24	3.18	2.72	2.92	2.70

^1^ According to FSC = 0.143.

## Data Availability

The cryo-EM maps of *Macrobrachium rosenbergii* nodavirus are available as EMD-9706 and EMD-9707 on the EMDB database and the corresponding atomic structures are deposited on Protein Data Bank with accession numbers 6JJC and 6JJD.

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
