# Peer review of "Sub-3 Å Cryo-EM Structures of Necrosis Virus Particles via the Use of Multipurpose TEM with Electron Counting Camera"

_ijms, 2021, doi:10.3390/ijms22136859_

Round 1
Reviewer 1 Report
The paper “3 Å cryo-EM structures of necrosis virus particles via the use of multi-purposed TEM With direct electron camera” advert for the use of more common electron microscopes for the use of high resolution structural studies. This is very useful as mentioned because the demand for high-end, dedicated, machines is high. I personally belief that the dedicated microscopes are optically not better than the multi-purpose microscopes, which limits the question here to the stability of the stage between autoloader and a side-entry stage. However, in this study the experimental for the different projects was not mentioned unfortunately. For the best microscope performance, the coherency of the beam is crucial as is clearly mentioned for the Talos but not for the Jeol 2100F Furthermore data on the FEG performance (e.g. coherency) is vital to make a fair comparison.
Stage drift and particle drift in ice upon exposure both limit the final resolution but have a different origin. Is there any data on stage drift between the stages (autoloader versus side entry stage)?
The image analysis done initially of the data from DGNNV is distracting because the same data was re-analyzed with more modern software and therefore does not contribute to the main focus of the paper, stage stability. Also the data from the Talos arctica was differently analyzed then the other data, I wonder if it is possible to use the same true out.
In the surge for cheaper options the side entry stage is clearly one option but there are other promising developments like the work from Henderson on 100 KeV microscopes. In the introduction
DeRosier and Crowther are mentioned but not the pioneering work of Joachim Frank on image analysis which I found somewhat disrespected.
Clearly the paper demonstrates once more that the key to the success of cryo-EM in structural biology is mainly the detector and the software that makes the difference and not so much the microscope anymore. The time that is needed to record images on low end microscopes will also be overcome with the availability of Serial EM which can be installed on every modern microscope and can record even faster than the software provided by the microscope manufacturer.
Some discussion on the real limiting factors for structural biology can be helpful as well
I will feel odd to place a very expensive camera on a similar prized microscope.
Author Response
General Replies
First of all, we are grateful the reviewers for spending valuable time on reading our manuscript title “3 Å cryo-EM structures of necrosis virus particles via the use of multi-purposed TEM with direct electron camera” and giving critical suggestions. By addressing these comments, we now substantially revise our manuscript by (1) expanding the Introduction and Discussion, (2) re-processing the all the data using advanced algorithms to further extend the resolutions so as to bring it to an acceptable level, we believe. Since we have furthered the resolution and suggest the ultimate limiting factor as the depth of field by 200 kV electrons, we now slightly modify the title “Sub-3 Å cryo-EM structures of necrosis virus particles via the use of multi-purposed TEM with electron counting camera “. In addition, we highlight the findings/achievements of this work as follows.
(1) 200 kV multi-purposed instruments enable structure determination of 30 nm GNNV virus particle to better than 3 Å resolution, the highest resolution for GNNV.
(2) 200 kV high-end instruments enable structure determination of 30 nm MrNV virus particles to better than 3 Å resolution, the highest resolution for MrNV.
(3) Comparison of instruments surprisingly shows multi-purposed instruments have similar stage characters as that of a high-end instrument.
(4) Resolution-limiting aberrations can now be corrected in silico by advanced algorithms.
(5) Our analysis shows the 2.7 Å resolution represents a ultimate limit posed on 30 nm by 200 kV electrons, which might be further lifted by Ewald sphere correction in the future.
Comments and Suggestions for Authors by Reviewer#1
The paper “3 Å cryo-EM structures of necrosis virus particles via the use of multi-purposed TEM With direct electron camera” advert for the use of more common electron microscopes for the use of high resolution structural studies. This is very useful as mentioned because the demand for high-end, dedicated, machines is high. I personally belief that the dedicated microscopes are optically not better than the multi-purpose microscopes, which limits the question here to the stability of the stage between autoloader and a side-entry stage. However, in this study the experimental for the different projects was not mentioned unfortunately. For the best microscope performance, the coherency of the beam is crucial as is clearly mentioned for the Talos but not for the Jeol 2100F Furthermore data on the FEG performance (e.g. coherency) is vital to make a fair comparison.
Stage drift and particle drift in ice upon exposure both limit the final resolution but have a different origin. Is there any data on stage drift between the stages (autoloader versus side entry stage)?
The image analysis done initially of the data from DGNNV is distracting because the same data was re-analyzed with more modern software and therefore does not contribute to the main focus of the paper, stage stability. Also the data from the Talos arctica was differently analyzed then the other data, I wonder if it is possible to use the same true out.
In the surge for cheaper options the side entry stage is clearly one option but there are other promising developments like the work from Henderson on 100 KeV microscopes. In the introduction
DeRosier and Crowther are mentioned but not the pioneering work of Joachim Frank on image analysis which I found somewhat disrespected.
Clearly the paper demonstrates once more that the key to the success of cryo-EM in structural biology is mainly the detector and the software that makes the difference and not so much the microscope anymore. The time that is needed to record images on low end microscopes will also be overcome with the availability of Serial EM which can be installed on every modern microscope and can record even faster than the software provided by the microscope manufacturer.
Some discussion on the real limiting factors for structural biology can be helpful as well
I will feel odd to place a very expensive camera on a similar prized microscope.
Replies to Reviewer#1
We are grateful for the efforts and comments of the reviewer. We are delighted that the reviewer reads our message that camera and algorithms are more crucial than microscopes despite that we had written the manuscript poorly. We address the limiting factors as suggested in the response to the comments. The responses are in point-by-point manner. In addition, we integrate our responses into our revised manuscript to enhance the significance and clarity of this work.
Comment 1_1: “However, in this study the experimental for the different projects was not mentioned unfortunately.
Reply 1_1: We now make tables to summarize the experiments for different projects. Table S1 for the imaging experiments on three experiments: setting A is DGNNV by JEOL 2100F-DDD), setting B is DGNV by FEI F20-K2, and setting C is MrNV by Talos-Falcon III. Table S2 and S3 for old and new processing.
Comment 1_2: “the coherency of the beam is crucial as is clearly mentioned for the Talos but not for the Jeol 2100F. Furthermore data on the FEG performance (e.g. coherency) is vital to make a fair comparison. ”
Reply 1_2: In the revised Discussion “Our tests on the two settings of multi-purposed TEMs revealed the results coming from the setting B with a counting camera are better than setting A by 0.4 Å, which is consistent with the observations made by Kayama et al. in comparing K2 and DE-20 [29]. However, we cannot entirely attribute the higher resolution to the superior DQE of the counting camera [26] since higher coherence was achieved on this setting (F20-K2) (Fig. S6) by the usage of a fairly small spot on the source to reduce the flux for minimizing the coincidence loss [34]. On the other hand, the coherence of the other setting (2100F-DE20) (Fig. S6) was reduced by using a larger spot, circumstantially selected in exchange for increased beam flux from the aging field emission gun. Besides, the spherical aberration on 2100 F (Cs 3.3 mm) is slightly poorer than F20 (Cs 2.3 nm) as the former was intended for the use of a phase plate [42] similar to the met by Kayama et al. [29].”
Please also see Fig. S6 for the comparison of the apparent coherence using the power spectra.
Comment 1_3: “Is there any data on stage drift between the stages (autoloader versus side entry stage)?“
Reply 1_3: Yes, please see Fig. S6 for stage drift of JEOL 2100F-Gatan 626, FEI F20-Gatan626, and Talos Arctica-autoloader. They are normalized using dose, not time; surprisingly they are quite similar.
Comment 1_4: “The image analysis done initially of the data from DGNNV is distracting because the same data was re-analyzed with more modern software and therefore does not contribute to the main focus of the paper, stage stability.“
Reply 1_4: We now present the results of old processing and new processing in the same section. The stage drift was taken care by global motion correction. However, there are two types of motion, stage and particle motions where global motion correction cannot eliminate the latter. The reprocessing invokes patch motion correction, close to particle motion correction since virus is much larger than common proteins. We believe the resolution extension by ~0.4 angstrom by reprocessing is a combination of correction of local motion correction and high-order aberrations. This paper not only addresses stage stability but aims to push the resolution to the limit to reveal the ceiling for 30 nm particle by 200 kV electrons.
Comment 1_5: “In the surge for cheaper options the side entry stage is clearly one option but there are other promising developments like the work from Henderson on 100 KeV microscopes.”
Reply 1_5: Thanks for this comment regarding the efforts Richard Henderson is making. We now put it in Discussion with perspectives of trend and limit “Recently, there is an increasing interest in using cryo-EM of lower accelerating voltage where the results from 200 kV cryo-EMs of high-end models start to rival with those from 300 kV instruments [22,24]. Of note is that the figure of 1.7 Å obtained for apo-ferritin by Wu et al. [22] may again merely reflect the resolution ceiling imposed by focus gradient on this 12 nm particle. Strikingly, this figure matches exactly the value calculated using the formula for Table 4 in Ref. 45. Were Ewald sphere correction to become a routine [46], one may expect similar results even attainable from a more affordable 100 kV instrument [49] given that a direct electron camera optimized for 100 kV electrons is available.“
Comment 1_6: “In the Introduction, DeRosier and Crowther are mentioned but not the pioneering work of Joachim Frank on image analysis which I found somewhat disrespected. “
Reply 1_6: Thanks for the crucial comment and we apologize for omitting Frank in the Introduction. We now include Frank by writing with historical perspectives “A much more difficult route for non-periodical objects was pioneered by Walter Hoppe [3], and eventually fulfilled by Joachim Frank in the establishment of the single particle reconstruction as a general method to include proteins with low or no symmetry [4], paving the way for the cryo-EM resolution breakthrough.“
Comment 1_7: “Clearly the paper demonstrates once more that the key to the success of cryo-EM in structural biology is mainly the detector and the software that makes the difference and not so much the microscope anymore. The time that is needed to record images on low end microscopes will also be overcome with the availability of Serial EM which can be installed on every modern microscope and can record even faster than the software provided by the microscope manufacturer. “
Reply 1_7: This is indeed so. Now we write in Discussion “Considering a high-end cryo-EM is beneficial for its high efficiency on data collection, it is now possible to use open-source microscope controlling software such as Leginon [50] or those related to Serial EM [51,52] for the automated data collection on multi-purposed TEMs to achieve similar efficiency. Besides, it is pertinent to use a new cryo-holder on multi-purposed TEMs that can keep the specimen at low temperature overnight without human attention. Those features for remedy have been either demonstrated or discussed in the work by Kayama et al [29]. “
Comment 1_8: “Some discussion on the real limiting factors for structural biology can be helpful as well “.
Reply 1_8: Thanks for this comment. We now write general limiting factors in Introduction “Since the resolution revolution of cryo-EM [11], nearly all high-resolution (< 3.5 Å) cryo-EM reconstructions including many virus capsid proteins reported to date were obtained using cryo-EMs operated at an accelerating voltage of 300 kV [12-17]. Those results are partly associated with a number of advantages provided by transmission electron microscopes (TEMs) with higher accelerating voltage. Those advantages include greater depth of field (less curvature for Ewald sphere) [18], larger mean-free path for electron, smaller phase errors induced by beam-tilt [19], and minimized specimen charging effect [20]. “ and in Discussion we reason the limitation we encounter with potential solutions provided “A question arises as to whether or not we do much better on these multi-purposed TEMs? Using a high-end cryo-EM of the same accelerating voltage (Talos Arctica, ThermoFisher) on a similar particle (MrNV), the same final resolution of 2.7 Å was ob-tained, suggesting 2.7 Å would represent a fundamental limit for 30 nm particles ulti-mately attainable from a 200 kV instrument. What would be then the limiting factors? As pointed out in a seminar paper by Zhang and Zhou [45] that has spurred instrument development for catalyzing the cryo-EM revolution, the resolution of a cryo-EM structure is a cumulative outcome from a number of factors. Those factors include the ones related to the specimen and others to the instrument such as stage stability and high-order aberrations caused by optics imperfection [19]. Our studies show the stage together with the specimen drifts on the multi-purposed TEMs were comparable to that of Talos Arctica and the resulting effect could be largely eliminated by applying motion correction to the movie data. To minimize the high-resolution phase errors induced by beam tilt and coma, the multi-purposed TEMs were both well-aligned for achieving parallel beam illumination [19] but not for the elimination of coma while additional coma-free alignment [19] was performed for the high-end 200 kV cryo-EM (Talos Arctica) using the provided software (ThermoFisher). If there were any residual errors, they were likely to have been corrected “in silico” [22] by the usage of the advanced software [38,39]. By ruling out those mentioned factors as the limiting factors, one may well ascribe the experimental limit of 2.7 Å, to a ceiling imposed by the focus gradient across the height of the specimen. Bearing this in mind, we found this figure of 2.7 Å coincided with the theoretical value calculated for a 30 nm particle using 200 kV electrons (see Table 4 in Ref. 45 and the reference therein). It is possible to use Ewald sphere correction [18, 45, 46] to lift this ceiling, which was demonstrated by a sub-2Å structure of adenovirus using a 300 kV instrument [47]. However, there is a rare case of a better-than-3.5Å resolution obtained for a 75 nm icosahedral particle without using the correction [48]. “.
Comment 1_9: “I will feel odd to place a very expensive camera on a similar prized microscope.”
Rely 1_9: This is why we could not justify getting a K2 on 2100F back in 2013. We instead got a DE-20. Now Okazaki NIPS got a refurbished K2 on 2100F (see Kayama et al [29]. We discuss this issue in Discussion “Since multi-purposed TEMs are much more affordable, the purchase and installation would meet much fewer financial obstacles. As a result, having a dedicated instrument in a laboratory of high-level biosafety would be appealing. It may seem odd to equip a low-priced TEM with a high-end counting camera almost as expensive as the TEM. Fortunately, the high-end counting cameras are now available in the refurbished form at a significantly reduced price.“
Thank you!

Reviewer 2 Report
The manuscript Wang and Chen is suppose to report the 3Å cryo-EM structures of necrosis virus particles with a 200Kv microscope. However, the aims of the authors is unclear, Betanodavirus structure has already been solved by Xray (Chen, NC et al . Crystal Structures of a Piscine Betanodavirus: Mechanisms of Capsid Assembly and Viral Infection. PLoS Pathog. 2015 and Cryo-EM (Ho, KL et al. Cryo-Electron Microscopy Structure of the Macrobrachium rosenbergii Nodavirus Capsid at 7Angstroms Resolution. Sci Rep 7, 2083 (2017). If the goal was to demonstrate that scientists with strong skill in cryo-em can achieve the same resolution than standard user with a 300Kev. This was also already know with for example work of Mengyu Wu et al. Sub-2 Angstrom resolution structure determination using single-particle cryo-EM at 200 keV. Journal of Structural Biology: X, 2020 or Merk, A et al. 1.8 Å resolution structure of β-galactosidase with a 200 kV CRYO ARM electron microscope (2020). IUCrJ 7, 639-643.
The work presented here have a very limited interest and Figure 4, cast doubt on the 3Å resolution claimed. Moreover, the text is unclear with many typographic errors and/ or orthography grammatical mistakes.
Some example below:
Lane 65 “of T = 3 GNNV” ???
Lane 87 “the software provided by the vendor” does it has a name?
Motion correction is obviously not visible in Figure 1A and is anyway never visible with a picture.
The explanation for the gain in resolution is unclear and Figure1 is odd. Why the special resolution is not shown?
The reviewer do not understand the purpose of Figure 2 panel D and E.
Lane 125: “we traveled to Stanford” I happy to know about it.
Author Response
General Replies
First of all, we are grateful the reviewers for spending valuable time on reading our manuscript title “3 Å cryo-EM structures of necrosis virus particles via the use of multi-purposed TEM with direct electron camera” and giving critical suggestions. By addressing these comments, we now substantially revise our manuscript by (1) expanding the Introduction and Discussion, (2) re-processing the all the data using advanced algorithms to further extend the resolutions so as to bring it to an acceptable level, we believe. Since we have furthered the resolution and suggest the ultimate limiting factor as the depth of field by 200 kV electrons, we now slightly modify the title “Sub-3 Å cryo-EM structures of necrosis virus particles via the use of multi-purposed TEM with electron counting camera “. In addition, we highlight the findings/achievements of this work as follows.
(1) 200 kV multi-purposed instruments enable structure determination of 30 nm GNNV virus particle to better than 3 Å resolution, the highest resolution for GNNV.
(2) 200 kV high-end instruments enable structure determination of 30 nm MrNV virus particles to better than 3 Å resolution, the highest resolution for MrNV.
(3) Comparison of instruments surprisingly shows multi-purposed instruments have similar stage characters as that of a high-end instrument.
(4) Resolution-limiting aberrations can now be corrected in silico by advanced algorithms.
(5) Our analysis shows the 2.7 Å resolution represents a ultimate limit posed on 30 nm by 200 kV electrons, which might be further lifted by Ewald sphere correction in the future.
Comments and Suggestions for Authors by Reviewer#2
The manuscript Wang and Chen is suppose to report the 3Å cryo-EM structures of necrosis virus particles with a 200Kv microscope. However, the aims of the authors is unclear, Betanodavirus structure has already been solved by Xray (Chen, NC et al . Crystal Structures of a Piscine Betanodavirus: Mechanisms of Capsid Assembly and Viral Infection. PLoS Pathog. 2015 and Cryo-EM (Ho, KL et al. Cryo-Electron Microscopy Structure of the Macrobrachium rosenbergii Nodavirus Capsid at 7Angstroms Resolution. Sci Rep 7, 2083 (2017). If the goal was to demonstrate that scientists with strong skill in cryo-em can achieve the same resolution than standard user with a 300Kev. This was also already know with for example work of Mengyu Wu et al. Sub-2 Angstrom resolution structure determination using single-particle cryo-EM at 200 keV. Journal of Structural Biology: X, 2020 or Merk, A et al. 1.8 Å resolution structure of β-galactosidase with a 200 kV CRYO ARM electron microscope (2020). IUCrJ 7, 639-643.
The work presented here have a very limited interest and Figure 4, cast doubt on the 3Å resolution claimed. Moreover, the text is unclear with many typographic errors and/ or orthography grammatical mistakes.
Some example below:
Lane 65 “of T = 3 GNNV” ???
Lane 87 “the software provided by the vendor” does it has a name?
Motion correction is obviously not visible in Figure 1A and is anyway never visible with a picture.
The explanation for the gain in resolution is unclear and Figure1 is odd. Why the special resolution is not shown?
The reviewer do not understand the purpose of Figure 2 panel D and E.
Lane 125: “we traveled to Stanford” I happy to know about it.
Replies to Reviewer#2
We are grateful for the efforts and valuable comments of the reviewer. We have substantially revised the manuscript accordingly and also completely re-processed the data. We believe our revision with enhanced clarity will expand the interest of our work. We now include a perspective on current advance using 200 kV to achieve sub-2A resolution structure and discuss the limitations on larger particles as in our case. We now highlight the main features of our work as follows, by which we believe this will be very interesting work now as opposed to the previous version that was written poorly.
(1) 200 kV multi-purposed instruments enable structure determination of 30 nm GNNV virus particle to better than 3 Å resolution, the highest resolution for GNNV.
(2) 200 kV high-end instruments enable structure determination of 30 nm MrNV virus particles to better than 3 Å resolution, the highest resolution for MrNV.
(3) Comparison of instruments surprisingly shows multi-purposed instruments have similar stage characters as that of a high-end instrument.
(4) Resolution-limiting aberrations can now be corrected in silico by advanced algorithms.
(5) Our analysis shows the 2.7 Å resolution represents a ultimate limit posed on 30 nm by 200 kV electrons, which might be further lifted by Ewald sphere correction in the future.
Comment 2_1: “However, the aims of the authors is unclear, Betanodavirus structure has already been solved by Xray (Chen, NC et al . Crystal Structures of a Piscine Betanodavirus: Mechanisms of Capsid Assembly and Viral Infection.”
Reply 2_1: We did manage to cite the work by Chen, NC et al. “Crystal Structures of a Piscine Betanodavirus: Mechanisms of Capsid Assembly and Viral Infection”, which was published in PloS Pathogens in 2015. We now explain our motivation clearly in Introduction “The first high-resolution structure of GNNV particles with triangular number of 3 (T=3, ca. 30 nm in diameter) was available from an X-ray crystal that diffracted to 3.6 Å [33], unfortunately falling short of crystallographic resolution. Therefore, one purpose of the present study is to test cryo-EM with the hope of generating a density map of better resolution so as to support the building of a more accurate atomic model with ease.“ We now also cite work by KL Ho et al. “Cryo-Electron Microscopy Structure of the Macrobrachium rosenbergii Nodavirus Capsid at 7Angstroms Resolution. Sci Rep 7, 2083 (2017)) in Introduction to explain our motivation and results “Since we were curious about the performance of a 200 kV automated cryo-EM installed in our center, we tested this cryo-EM (Talos Arctica) with particles of a shrimp nodavirus (MrNV) (T=3, ca. 30 nm in diameter), evolutionarily related to GNNV [40], and also obtained a structure of with a resolution of 2.7 Å, updating the 7 Å cryo-EM structure obtained earlier by a 200 kV instrument [41] similar to one of the settings tested here.“.
Comment 2_2: “Figure 4, cast doubt on the 3Å resolution claimed.”
Reply 2_2: We now update the maps in Figure 4 to match the claimed resolutions.
Comment 2_3: “Lane 65. of T = 3 GNNV” ???.”
Reply 2_3: We now re-write it as GNNV particles with triangular number = 3 (T =3).
Comment 2_4: “Lane 87. the software provided by the vendor” does it has a name?”
Reply 2_4: This motion correction python code was provided by DDD, Ltd. as “For individual micrograph, the drift for each frame was analyzed globally and corrected by using the DE_processframe.py software from the vendor (DDD, Ltd.). “ in the revision.
Comment 2_5: “Motion correction is obviously not visible in Figure 1A and is anyway never visible with a picture.”
Reply 2_5: Please see particle contrast before and after motion correction in Figure 1A, and also the power spectra before and after the motion correction in Fig. S1A & B.
Comment 2_6: “The explanation for the gain in resolution is unclear and Figure1 is odd. Why the special resolution is not shown?”
Reply 2_6: We now write in Results 2.1 “Initially, even with motion correction (Fig. S2) the overall resolution was stalled at 4.21 Å with accumulated dose of accumulated dose of ~30 e-/Å2. To improve the resolution, we investigated the radiation damage effect by dose fractionation analysis using frame exclusion [34]”, and “As shown in Figure 1C, the best overall resolution of 3.79 Å was achieved by limting the total dose to ~14 e-/Å2 “. We now move the power spectra to Fig. S1 with the addition of a circle to show the spatial resolution of cut-off in the spectra.
Comment 2_7: “The reviewer do not understand the purpose of Figure 2 panel D and E.”
Reply 2_7: The purpose of Figure 2D that shows side-chain is to support the resolution claim. This is a common practice: please see Figure 1 by Kayama et al. in [29] and Figure 2 in [48]. In particular, at this level of resolution, the difference of Phe and Tyr can be discerned due to the hydroxyl group of the latter (Figure 2D). Furthermore, it is a shared view in the cryo-EM community that ordered water molecules can be observed at the resolution of ~2.7 Å (see Grba DN, Hirst J. Mitochondrial complex I structure reveals ordered water molecules for catalysis and proton translocation, NSMB 2020, 27, 892-900. https://doi:10.1038/s41594-020-0473-x
Comment 2_8 “If the goal was to demonstrate that scientists with strong skill in cryo-em can achieve the same resolution than standard user with a 300Kev. This was also already know with for example work of Mengyu Wu et al. Sub-2 Angstrom resolution structure determination using single-particle cryo-EM at 200 keV. Journal of Structural Biology: X, 2020 or Merk, A et al. 1.8 Å resolution structure of β-galactosidase with a 200 kV CRYO ARM electron microscope (2020). IUCrJ 7, 639-643.”
Reply 2_8: We now put in Introduction “Consequentially, there is a growing interest in using cryo-EMs with lower accelerating voltage that are more affordable for structure determination. The recent work by Herzik and Lander [22] on breaking the boundary of 2 Å using apo-ferritin, a conformationally homogeneous test specimen of 12 nm [23], has again demonstrated the power of a 200 kV instrument. Remarkably, using a newer model of 200 kV cryo-EM with a superior field-emission source, Subramaniama group obtained a 1.8 Å structure for beta-galactosidase [24]. Since this specimen is a more challenging sample than apo-ferritin due to its lower symmetry, this work truly demonstrates a state-of-the-art 200 kV instrument can rival with the existing high-end 300 kV machine….. It should be pointed out that it has remained largely unknown whether or not the resolution achieved for apo-ferritin (ca. 12 nm) on a 200 kV multi-purposed instrument can be similarly obtained for much larger particles on a similar setting. To evaluate the potential performance of a multi-purposed TEM on virus structure determination, we embarked on virus particles of a betanodavirus with 30 nm in size that can infect the grouper fish [30] “ and in Discussion “By ruling out those mentioned factors as the limiting factors, one may well ascribe the experimental limit of 2.7 Å, to a ceiling imposed by the focus gradient across the height of the specimen. Bearing this in mind, we found this figure of 2.7 Å coincided with the theoretical value calculated for a 30 nm particle using 200 kV electrons (see Table 4 in Ref. 45 and the reference therein)……Recently, there is an increasing interest in using cryo-EM of lower accelerating voltage where the results from 200 kV cryo-EMs of high-end models start to rival with those from 300 kV instruments [22,24]. Of note is that the figure of 1.7 Å obtained for apo-ferritin by Wu et al. [22] may again merely reflect the resolution ceiling imposed by focus gradient on this 12 nm particle as this figure matches exactly the value calculated using the formula for Table 4 in Ref. 45. “
Thank you!

Round 2
Reviewer 1 Report
The paper now gives a nice balanced overview of the different microscope settings and stages. I hope scientists dare to put dedicated cameras on allround microscopes and that in the future some nice structures are produced this way. This paper can be a good start to get some movement in the TEM application field.
Author Response
We are very grateful for the supporting comments and share the same wish that scientists would take this approach.

Reviewer 2 Report
The manuscript improved, but is still not convincing enough
1-As previously asked it would be good to have the spatial or local resolution colouring in Fig1B.
2-The Figure 2D does not make sense and should be removed. Cryo-em is not like X-Ray, it is not because the Valine 104 is well defined that other ones will be. As an example, for the SARS-CoV-2 Spike, almost of map go down to 2A for the central core, but most of the map have a resolution of 5-6A for the RBD because this part of the molecule is flexible. Hence, in addition to the general resolution, as asked in point 1and shown for all cryo-em map the local resolution is essential.
3-Figure 4 at 3A you should not see side chain out of the electron density. As we can see for R50, T46 and T35.
It would also be good to know how model refinement was done and have the CC for the map and pdb model.
Author Response
We are thankful for the reviewer’s comments and keen insights. We believe any healthy doubt would only help improve the manuscript. We therefore substantially revise our manuscript accordingly and address your comments in a point-by-point manner as in the attached PDF file of "Response to Reviewer#2 Comments".
